# A Novel Anti-CD73 Antibody That Selectively Inhibits Membrane CD73 Shows Antitumor Activity and Induces Tumor Immune Escape

**DOI:** 10.3390/biomedicines10040825

**Published:** 2022-03-31

**Authors:** Markus Kellner, Bettina von Neubeck, Bastian Czogalla, Regina Feederle, Binje Vick, Irmela Jeremias, Reinhard Zeidler

**Affiliations:** 1Research Group Therapeutic Antibodies, Helmholtz Zentrum München German Research Center for Environmental Health, Feodor-Lynen-Str. 21, 81377 Munich, Germany; markus.kellner@helmholtz-muenchen.de (M.K.); bettina.neubeck@gmx.de (B.v.N.); 2Department of Obstetrics and Gynecology, University Hospital, Ludwig Maximilian Universität, 81377 Munich, Germany; bastian.czogalla@med.uni-muenchen.de; 3Core Facility Monoclonal Antibodies, Helmholtz Zentrum München German Research Center for Environmental Health, Ingolstädter Landstraße 1, 85764 Neuherberg, Germany; regina.feederle@helmholtz-muenchen.de; 4Research Unit Apoptosis in Hematopoietic Stem Cells, Helmholtz Zentrum München German Research Center for Environmental Health, Feodor-Lynen-Str. 21, 81377 Munich, Germany; binje.vick@helmholtz-muenchen.de (B.V.); irmela.jeremias@helmholtz-muenchen.de (I.J.); 5Department of Pediatrics, University Hospital, Ludwig Maximilian Universität, 80337 Munich, Germany; 6Department of Otorhinolaryngology, University Hospital, Ludwig Maximilian Universität, 81377 Munich, Germany

**Keywords:** CD73, adenosine, immune evasion, extracellular vesicles, cancer therapy, therapeutic antibody

## Abstract

CD73 catalyzes the conversion of ATP to adenosine, which is involved in various physiological and pathological processes, including tumor immune escape. Because CD73 expression and activity are particularly high on cancer cells and contribute to the immunosuppressive properties of the tumor environment, it is considered an attractive target molecule for specific cancer therapies. In line, several studies demonstrated that CD73 inhibition has a significant antitumor effect. However, complete blocking of CD73 activity can evoke autoimmune phenomena and adverse side effects. We developed a CD73-specific antibody, 22E6, that specifically inhibits the enzymatic activity of membrane-tethered CD73 present in high concentrations on cancer cells and cancer cell-derived extracellular vesicles but has no inhibitory effect on soluble CD73. Inhibition of CD73 on tumor cells with 22E6 resulted in multiple effects on tumor cells in vitro, including increased apoptosis and interference with chemoresistance. Intriguingly, in a xenograft mouse model of acute lymphocytic leukemia (ALL), 22E6 treatment resulted in an initial tumor growth delay in some animals, followed by a complete loss of CD73 expression on ALL cells in all 22E6 treated animals, indicating tumor immune escape. Taken together, 22E6 shows great potential for cancer therapy, favorably in combination with other drugs.

## 1. Introduction

CD39 and CD73 (5′-ectonucleotidase, NT5E) are key ectoenzymes in the generation of anti-inflammatory adenosine (ADO) by dephosphorylating adenosine triphosphate (ATP) to ADP to AMP to ADO in a stepwise reaction. CD73 is a GPI-linked membrane-tethered glycoprotein and the rate-limiting enzyme that catalyzes the last step in this process, making it the principal enzyme to produce extracellular ADO, which plays an important role in many physiological processes [1]. Consequently, CD73 is expressed on several healthy tissues either constitutively or upon activation [2] where it predominantly elicits anti-inflammatory signals upon binding to ADO receptors on, e.g., activated immune effector cells [3].

CD73 is also regularly expressed at high levels in various types of cancer, including carcinoma of different localizations, melanomas, and lymphomas [3,4]. Expression of CD73 is induced by hypoxia and by inflammatory mediators such as TGF-β [3] so that in the tumor microenvironment, its expression is not confined to cancer cells but is also detectable on tumor-infiltrating immune effectors, such as macrophages, dendritic cells, and lymphocytes [3,5]. Thus, the enzyme accounts for the particularly high local concentrations of ADO that contribute to the immunosuppressive properties of the tumor environment and hence to tumor progression and immune escape [6]. Moreover, high expression levels of CD73 on cancer cells are also associated with chemoresistance [4,7,8], enhanced migration and invasion [9], metastatic spread [10], and enhanced tumor angiogenesis [11]. Consequently, CD73 expression is associated with a poor clinical prognosis in some tumor entities [12,13,14,15] and is therefore considered a relevant and highly attractive therapeutic target. In line, several CD73 blocking antibodies and small molecules are currently under clinical development for the treatment of cancer, mostly as part of combination therapies (see, e.g., www.clinicaltrials.gov, accessed on 21 February 2022).

Here, we describe a new CD73 blocking monoclonal antibody, 22E6, which effectively inhibits the enzymatic activity of the membrane-tethered form of human CD73 (mCD73) on cancer cells and tumor-derived extracellular vesicles at nanomolar concentrations. We show that 22E6 blocks CD73 in a non-competitive manner, that it directly induces apoptosis in triple-negative breast cancer cells in vitro, and that it has a transient antitumor effect on patient-derived acute lymphatic leukemia (ALL) cells in vivo. Intriguingly, in a xenograft mouse model of acute lymphatic leukemia (ALL), 22E6 treatment of mice carrying CD73+ ALL cells resulted in a downregulation of CD73 expression. This observation demonstrates, on the one hand, that CD73+ cancer cells are sensitive to 22E6 treatment but, on the other hand, corroborates the clinical problem of resistant immune escape variants induced by cancer monotherapies. In essence, we show that 22E6 is a promising novel proprietary antibody candidate as part of combination therapies.

## 2. Materials and Methods

### 2.1. Cells and Culture Conditions

MDA-MB231 cells were kindly provided by Dr. Chiara Riganti (Department of Oncology, University of Turin, Torino, Italy). MDA-MB231, U138 MG, and T47-D cells were purchased from ATCC and were cultured in DMEM/F12 medium (Gibco BRL, Karlsruhe, Germany). GBM20 cells were a kind gift from Prof. R. Glass (Munich, Germany) and were kept in DMEM/F12 medium supplemented with 1× B 27, 1% *v*/*v* penicillin-streptomycin, 10 ng/mL human EGF (R&D Systems, Minneapolis, MN, USA) and 10 ng/mL human FGF-basic (PeproTech, Hamburg, Germany). All other cell lines were grown in RPMI 1640 medium supplemented with 8 % *v*/*v* fetal bovine serum, 1% *v*/*v* L-Glutamine, and 1 % *v*/*v* penicillin-streptomycin). All cells were incubated in a humidified CO_2_ incubator at 37 °C, 18.3% O_2_ and 5% CO_2_. Cell line identification was performed by STR-PCR (Eurofins, Ebersberg, Germany).

### 2.2. Transfection of 293T Cells

The 293T cells were transfected with an expression plasmid, coding for human CD73 (human CD73 ORF cDNA clone expression plasmid, HIS-tagged, (Sino Biologicals, Beijing, China) using LipofectamineTM 2000 transfection reagent (ThermoFisher Scientific, Menzel, Germany) according to manufacturer’s instructions. Cells were selected for transfected cells with 80 µg/mL Hygromycin C.

### 2.3. Isolation of Peripheral Blood Mononuclear Cells (PBMCs)

30 mL of fresh heparinized blood from healthy donors were diluted with PBS and loaded slowly on top of a layer of 10 mL of Pancoll (Pan Biotech, Aidenbach, Germany). The tubes were centrifuged at 800× *g* for 25 min. PBMCs were carefully collected from the interface between Pancoll and plasma and washed three times in PBS. Cells were immediately used for experiments or cryopreserved for later use.

### 2.4. Flow Cytometry

For analysis of surface CD73 expression by flow cytometry, 100,000 cells were stained with 22E6 hybridoma supernatant diluted 1:2 in FACS-buffer (PBS + 2% FSC) or with hybridoma supernatant of an appropriate Ig2a isotype control antibody for 20 min followed by staining with an anti-rat secondary antibody labeled with Alexa Fluor^®^ 647 (Jackson Immuno Research). All stainings were performed on ice.

### 2.5. Cloning and Production of Fab-22E6

Variable immunoglobulin sequences (H+L) of the 22E6 antibody were obtained by rapid amplification of cDNA ends (RACE; ThermoFisher). For this, original rat-hybridoma cell lines were harvested, and the total RNA was isolated by phenol/chloroform extraction and isopropanol precipitation. Subsequently, RACE was performed using Ig isotype-specific primers for cDNA synthesis (Appendix A). After cDNA purification by S.N.A.P. columns, a polyC-tail was added to the strands by a terminal deoxyribonucleotidyl-transferase (TdT). Finally, a PCR was conducted on the cDNA template to amplify the variable immunoglobulin sequences. In silico an IL-2 secretion signal was added 5′ and the human kappa constant part (light chain) or the human IgG1 constant Fab part (Heavy chain) were added 3′, respectively, to the PCR amplified variable sequences, and this construct was synthesized (GenScript, Leiden, The Netherlands). The sequences were cloned into a pcDNA3 vector and transfected into HEK293 cells. Fab22E6 was purified from the culture supernatant by applying it to a CaptureSelect KappaXL column (ThermoFisher). The elution was performed by a mild pH buffer (pH = 5.0) as described in the manufacturer’s manual (40% (*v*/*v*) propylene glycol, 20 mM sodium acetate, 1.0 M MgCl_2_). The Fab was eventually dialyzed against PBS and concentrated with Amicon-Ultra centrifugation filter units (Merck, Darmstadt, Germany).

### 2.6. Western Blotting Analysis

For analysis of CD73 expression by western blotting, whole-cell lysates were prepared. The cell pellet was washed in PBS, resuspended in RIPA buffer with protease inhibitors (Roche, Penzberg, Germany) and incubated on ice for 20 min. The lysate was centrifuged for 20 min at 4 °C at 15,000× *g* rpm to precipitate cell fragments. The supernatant was transferred to a new tube, and the protein content was determined with a Bradford assay (Bio-Rad). 20 µg of cell lysate were on an SDS-PAGE, electroblotted on a nitrocellulose membrane, and blocked with 5% nonfat dry milk in TBST. Following primary antibodies were used for Western blotting: anti-CD73 (rabbit anti-human, Abcam), anti-CD63 (rat anti-human, produced at the monoclonal antibody core facility, Helmholtz Center Munich), anti-Calnexin (mouse anti-human, BD Bioscience), anti-α-tubulin (mouse anti-human, GeneTex, Irvine, CA, USA), anti-TSG-101 (mouse anti-human, GeneTex), anti-CD81 (mouse anti-human, BioLegend, Koblenz, Germany). All primary antibodies were incubated at 4 °C overnight. The membrane was washed and incubated with appropriate HRP-conjugated secondary antibodies at room temperature for 2 h. The following secondary antibodies were used: anti-rat IgG-HRP (Cell Signaling), anti-mouse IgG-HRP (Cell Signaling Technology, Frankfurt a. Main, Germany), and anti-rabbit IgG-HRP (Cell Signaling). Detection of proteins was performed with the ECL system (GE Healthcare, Freiburg, Germany).

### 2.7. Immunoprecipitation

Immunoprecipitation was performed using CNBr-activated Sepharose 4 Fast Flow beads (GE Healthcare). 1 g beads were resuspended in 10 mL of 1 mM HCl and incubated at room temperature for 20 min. Beads were pelleted for 1 min at 3000× *g* and washed 15 times. Then, a mouse anti-rat IgG2a antibody was coupled to the beads (2 mg antibody in coupling buffer with 0.3 M NaHCO_3_, 1.5 M NaCl, pH 8.3) at room temperature for 1 h. Beads were washed in coupling buffer, and all remaining binding sites were blocked with ethanolamine (1 M) at room temperature for 2 h. After washing in washing buffer (0.1 M Tris/HCl, 0.5 M NaCl, pH 4.0) and in NaOAc buffer (0.1 M NaOAc, 0.5 M NaCl), beads were resuspended in PBS and used for coupling with 22E6. Therefore, 500 µL 22E6 hybridoma supernatant was incubated with 60 µL anti-subclass specific beads overnight at 4 °C. Beads were then washed in PBS and incubated with 2 mg cell lysate or 0.75 µg recombinant soluble His-tagged human CD73 (Sino Biologicals) at 4 °C overnight. Beads were washed in RIPA buffer with protease inhibitors, PBS+ (PBS, 0.5 % N-Laurylsarcosine, 0.1 % SDS). The supernatant was discarded, beads were resuspended in 3× Laemmli buffer, and incubated at room temperature for 3 min. After centrifugation for 5 min at 1000× *g,* the supernatant was loaded onto an SDS-PAGE.

### 2.8. Isolation of Tumor-Derived Extracellular Vesicles (TEVs)

Extracellular vesicles (EVs) were purified from cell culture supernatants or from primary ascites by serial centrifugation. Briefly, conditioned cell culture supernatants from GBM20 cells or ascites were centrifuged for 10 min at 300× *g* and 20 min at 5000× *g* to deplete cells and cellular debris. The supernatant was then filtered through a 0.8 µm PES filter and concentrated by ultracentrifugation at 100,000× *g* at 4 °C for 2 h. The concentrated EVs were washed once in phosphate-free HEPES buffer (2 mM MgCl_2_, 120 mM NaCl, 5 mM KCl, 20 mM Hepes, pH 7.4) and pelleted again by ultracentrifugation. EVs were further purified by floating into a discontinuous OptiPrep^TM^ gradient (Sigma-Aldrich, Deisenhofen, Germany) by ultracentrifugation at 4 °C and 160,000× *g* for 4.5 h. Eight fractions of 500 µL each were collected from the gradient. Dot blots were performed to analyze all fractions for the presence of the EV markers CD63, CD81, TSG-101, as well as for CD73. EVs were quantified by nano-tracking particle analysis (NTA) as described elsewhere [16]. Furthermore, the protein concentration and density of each fraction was determined. EV containing fractions (usually fractions 2 and 3) were pooled and washed and resuspended in HEPES buffer. The final EV pellet was used for WB analysis or CD73-activity assays.

### 2.9. Generation of the 22E6 Antibody

For the generation of antibody 22E6, Wistar rats were immunized with TEVs isolated from glioblastoma cell line U138 MG. Fusion of spleen cells with the myeloma cell line P3x63.653 was performed according to standard procedures for the generation of monoclonal antibodies. IgG-positive supernatants were screened for binding to CD73-positive cells by flow cytometry. Clone 22E6 is of isotype IgG2a/kappa and was subcloned twice by limiting dilution to obtain a stable monoclonal hybridoma cell line.

### 2.10. CD73 Activity Assay

The activity of CD73 can be determined by the analysis of AMP hydrolysis yielding free inorganic phosphate in a reaction mixture. For this, cells were grown to approximately 70% confluency in a 24-well plate. Adherent cells were washed three times with phosphate-free glucose buffer (2 mM MgCl_2_, 120 mM NaCl, 5 mM KCl, 10 mM glucose, 20 mM Hepes, pH 7.4). Cells were incubated with 200 µL glucose buffer per well, supplemented with 1 mM AMP (reaction buffer), at 37 °C for 30 min followed by the quantification of free phosphate. To study inhibition, cells were pre-incubated with APCP (Adenosine 5′-(α,β-methylene)diphosphate, Sigma Aldrich) or 22E6 at 37 °C for 30 min before addition of AMP (Sigma-Aldrich). To determine the amount of free phosphate, 100 µL of the reaction mix were transferred into a 96-flat-bottom well plate. 50 µL 0,5 M H_2_SO_4_ and 50 µL 0.4 % ammonium molybdate, freshly supplemented with 10 % (*w*/*v*) ascorbic acid), were added. The reaction was incubated for 30 min at room temperature in the dark. The amount of free phosphate was analyzed photometrically at 820 nm.

To measure the activity of soluble CD73 (sCD73), 80 ng recombinant human CD73 (Sino Biologicals) in glucose buffer was pre-incubated for 30 min APCP or 22E6 as described above. Then, AMP was added at a final concentration of 100 µM, and the reaction was incubated at 37 °C for 1 h. The quantification of free phosphate was performed using the Malachite Green Phosphate Assay Kit (Sigma-Aldrich), following the manufacturer’s instructions.

### 2.11. Apoptosis Assay and Analysis of Mitochondrial Activity

The analysis of apoptotic cells was performed by AnnexinV-Cy5 staining of cells according to the manufacturer’s instructions (BioVision, Ilmenau, Germany). Briefly, cells were harvested, pelleted, and resuspended in 1× Annexin V binding buffer. Cells were stained with AnnexinV-Cy5 (1:500) at room temperature for 5 min, washed, and analyzed by flow cytometry. To measure mitochondrial activity, cells were incubated with 25 nM TMRE (tetramethylrhodamine ethyl ester; Abcam) at 37 °C for 30 min in a CO_2_ incubator before detachment and Annexin V staining. TMRE was detected in the PE channel by flow cytometry.

### 2.12. In Vivo Treatment Trials

All animal trials were performed in accordance with the current ethical standards of the official committee on animal experimentation (written approval by Regierung von Oberbayern, tierversuche@reg-ob.bayern.de; ROB-55.2Vet-2532.Vet_02-16-7 and ROB-55.2Vet-2532.Vet_03-16-56). Primary patient ALL cells were engrafted and serially passaged in NOD.Cg-Prkdc^scid^ Il2r^tm1Wjl^/SzJ (NSG) mice (The Jackson Laboratory, Bar Harbour, ME, USA) as reported recently [17], to generate the patient-derived xenograft (PDX) cells. PDX cells were lentivirally modified to express enhanced firefly luciferase for sensitive and repetitive monitoring of tumor burden by bioluminescence in vivo imaging (BLI). Cryopreserved PDX cells were used to determine CD73 expression to select samples appropriate for an in vivo therapy trial. For in vivo therapy trials, PDX cells of ALL 272 or ALL 1124 were thawed and injected into groups of NSG mice (1.7 × 10^6^ ALL 272 cells or 5 × 10^6^ ALL 1124 cells per mouse). Three to 14 days post-injection, mice were treated with 22E6 (100 µg per application) or left untreated. Tumor outgrowth was repeatedly monitored and quantified by BLI. When BLI signals reached a total flux of 1 × 10^11^ Photons/second, or if mice showed clinical signs of illness, mice were sacrificed, and PDX cells were reisolated from murine bone marrow. Mice dying in inhalation anesthesia were excluded from the reanalysis. Data from two independent experiments were integrated.

### 2.13. Statistics

Statistics were calculated with GraphPad Prism applying the Student’s *t*-test. * *p* < 0.05; ** *p* < 0.01; *** *p* < 0.001; **** *p* < 0.0001. Error bars represent the SD, and the number of replicates was at least *n* ≥ 3 throughout the experiments. Interventional studies involving animals or humans, and other studies that require ethical approval, must list the authority that provided approval and the corresponding ethical approval code.

## 3. Results

### 3.1. 22E6 Specifically Binds CD73 on Human Cancer Cell Lines

The antibody 22E6 was generated by immunizing rats with extracellular vesicles derived from the human glioblastoma cells line U138 MG which expresses high levels of membrane CD73 (Figure 1C). Hybridoma obtained from this immunization were screened for CD73-specific antibodies by flow cytometry using CD73+ subclone of HEK293 cells obtained by the stable transfection of a CD73 expression plasmid, and CD73-negative parental HEK293 cells as a control. The antibody 22E6 bound to transfected but not to parental HEK293 cells (Figure 1A). To further verify its CD73 specificity, we performed immunoprecipitation with 22E6 and lysates from U138 MG cells, followed by an immunoblot hybridized against a commercial anti-CD73 antibody. Clearly, 22E6 but not an isotype control antibody, efficiently precipitated CD73 from U138 MG lysates (Figure 1B). We next investigated by flow cytometry the binding of 22E6 to a panel of permanent cancer cell lines and cells isolated from primary malignant ascites from patients with ovarian cancer. This experiment revealed that most of the tested cell lines and primary ascites-derived cells stained positive with 22E6 (Figure 1C), while T47-D breast cancer cells were completely negative (Figure 1C). These results were confirmed by immunoblot analyses on cell lysates using a commercial CD73 antibody.

### 3.2. 22E6 Inhibits Membrane CD73 on Tumor Cell Lines

Because CD73 is an enzyme that, for example, catalyzes the generation of ADO that may have immunosuppressive properties at high concentrations, we next evaluated whether 22E6 inhibits its enzymatic function. For this, we incubated human U138 MG cells in a phosphate-free glucose buffer containing the CD73 substrate AMP (final concentration 1 mM) for 30 min. The competitive CD73 small molecule inhibitor α,β-methylene-ADP (APCP; final concentration 10 µM) [18], and an isotype antibody were used as the positive and negative control, respectively. We then measured CD73 activity by quantifying the concentration of free inorganic phosphate, which is a product of CD73-catalyzed AMP hydrolysis. As depicted in Figure 2A, the activity of CD73 on U138 MG cells was significantly reduced by 22E6. The IC50 was measured by using a serial dilution of 22E6 (final concentration 10 µg/mL to 0.1 ng/mL) and was determined as 0.5 µg/mL (i.e., approx. 3.5 nM) (Figure 2B). It was seen that 22E6 inhibited CD73 activity on a panel of cancer cell lines (of which two are shown here) while having no effect on CD73-negative T47-D cells as expected (Figure 2C). Notably, in contrast to APCP, 22E6 did not inhibit CD73 activity completely, even at high concentrations, for unknown reasons (Figure 2A).

### 3.3. 22E6 Inhibits CD73 Enzymatic Activity on Tumor Derived Extracellular Vesicles

Tumor-derived extracellular vesicles (TEVs) from CD73-positive cancer cells have been described to carry functional CD73 and suppress T-cell function through adenosine production [19]. TEVs from a glioblastoma cell line (GBM20) were purified by an OptiPrep^TM^ Iodixanol density gradient. All fractions were analyzed by dot blots for CD73 and vesicles markers CD63, CD81, and TSG-101 (Figure 3A). Furthermore, the vesicle number was measured using a ZetaView tracking analyzer (Particle Metrix, Inning am Ammersee, Germany), and the protein content was determined with a Bradford assay (Figure 3B). These results revealed enrichment of CD73-positive vesicles in fraction 3 of the gradient. This fraction was washed and re-analyzed for CD73 and for vesicles markers CD63 and CD81 by western blot. Calnexin served as a negative control to exclude cellular contaminations of the vesicle preparation (Figure 3C). In the next step, we analyzed CD73 activity on these vesicles as described above. As it turned out, TEVs cargo active CD73 and, more importantly, CD73-activity could also be reduced by 22E6 (Figure 3D).

Additionally, we analyzed TEVs from cancer patient material. Therefore, TEVs were purified by density gradient from ovarian cancer patient-derived ascites and analyzed in detail for CD73 and vesicle markers as described above (Appendix A). Also patient-derived TEVs carried active CD73, which could also be inhibited by 22E6 (Figure 3E). To further investigate the presumed tumor-promoting role of CD73-positive TEVs, we checked whether the CD73 activity, and thus, the possibility to generate ADO, can be transferred to CD73-negative cells by TEVs. To this end, we incubated CD73-negative T47-D cells with CD73-positive TEVs for 48 h. We then washed the cells to remove unbound TEVs and measured the CD73 activity. The result showed that CD73 activity could be detected on CD73-negative cells after pre-incubation with TEVs (Figure 3F). These results emphasize the role of CD73-TEVs in the creation of ADO, which in turn may contribute to an immunosuppressive tumor environment, and that 22E6 could help to relieve this ADO-mediated effect.

### 3.4. 22E6 Is a Non-Competitive CD73 Inhibitor

Membrane CD73 is a homodimeric ectoenzyme that is tethered to the plasma membrane via a glycosylphosphatidylinositol (GPI) anchor (mCD73). CD73 consists of two domains, which undergo conformational changes during the enzymatic reaction [3,20]. To elucidate the mode of CD73 inhibition by 22E6, we incubated U138 MG cells with 22E6 in the presence of increasing concentrations of AMP. The kinetics of CD73 activity was analyzed and compared to the competitive inhibitor APCP (Figure 4A). As expected for a competitive inhibitor, APCP did not decrease Vmax (3.16 µM Pi/min vs. 4.2 µM Pi/min) but rather increased Km (125.3 µM vs. 665.3 µM). In contrast, 22E6 decreased Vmax (3.16 µM Pi/min vs. 2 µM Pi/min) while having no effect on Km (125.3 µM vs. 133.7 µM). These data suggest that 22E6 does not compete with the substrate AMP for binding to the active center of the enzyme.

To further analyze the mode of inhibition by 22E6, we produced the Fab fragment of 22E6 as described in Materials and Methods. We first successfully proved its binding to CD73-positive U138 MG cells by flow cytometry (Appendix A) and then analyzed its inhibitory effect. Interestingly, in contrast to the complete 22E6 antibody, the 22E6 Fab fragment did not detectably inhibit CD73 activity (Figure 4B). This result confirms our hypothesis that 22E6 does not directly inhibit CD73 activity by competitively blocking the active center but rather inhibits the enzyme in a non-competitive manner or by steric inhibition.

### 3.5. 22E6 Does Not Inhibit the Enzymatic Activity of Soluble CD73 (sCD73)

Because CD73 also exists in a soluble form (sCD73), we next analyzed its interaction with 22E6 in more detail. For this, we first performed immunoprecipitation of sCD73 with 22E6 coupled to Sepharose beads and analyzed the precipitates with immunoblot using a commercial CD73 antibody. The result revealed that 22E6 efficiently precipitated sCD73 (Figure 5A). In order to investigate the inhibition of sCD73 by 22E6, we next pre-incubated 80 ng of recombinant sCD73 with APCP (10 µM), 22E6, or an isotype control antibody (each used at a final concentration of 5 µg/mL) at room temperature for 30 min before adding AMP (100 µM final concentration), incubating at 37 °C for another 60 min, and measuring the inorganic phosphate with a Malachite Green assay. Interestingly, and in sharp contrast to APCP, 22E6 did not detectably interfere with sCD73 activity (Figure 5B).

### 3.6. 22E6 Induces Apoptosis in TNBC

CD73 can enhance the chemoresistance of tumor cells [4,7,21,22] and is a negative prognostic marker for triple-negative breast cancer (TNBC) [4]. Therefore, we examined the effect of 22E6 alone or in combination with the anthracycline Doxorubicin (DOX) for the induction of apoptosis in TNBC cells. For this, we incubated MDA-MB231 cells with 22E6 (5 µg/mL) alone or in combination with DOX (250 nM) and measured apoptosis with an Annexin assay by flow cytometry. APCP and an isotype antibody were used as controls.

Interestingly, in contrast to both APCP and the isotype antibody, 22E6 alone induced apoptosis and also had a significant additive effect with DOX (Figure 6A). Analysis of the metabolic activity by TMRE staining showed that 22E6 reduced mitochondrial activity alone and particularly in combination with DOX (Figure 6B). Again, the small molecule inhibitor APCP did not show this effect. In addition, neither 22E6 nor APCP induced apoptosis in CD73-negative 293T cells.

### 3.7. 22E6 Selects for a CD7^3dim/neg^ Subpopulation of Patient Derived ALL Cells In Vivo

To analyze the potential anti-tumor effect of 22E6 in vivo, we performed an experiment in immunocompromised NSG mice transplanted with patient-derived xenograft (PDX) acute lymphatic leukemia (ALL) cells. First, we screened several PDX samples derived from different ALL patients for CD73 surface expression and enzymatic activity as described above. We identified two samples, ALL272 and ALL1124, that both showed decent CD73 surface expression and enzyme activity (Figure 7A,B).

We transplanted ALL272 or ALL1124 cells into mice and treated them either with 22E6 (100 µg/animal, once per week) or left them untreated. Tumor burden was repeatedly monitored by in vivo bioluminescence imaging (BLI). The 22E6 treatment had only a minor effect on the growth of transplanted ALL272 cells, while measurements on days 35 and 42 after the beginning of treatment revealed a significant growth delay of ALL-1124 in mice treated with 22E6 (Figure 7C).

Cancer immune escape is a major clinical problem and an explanation for the development of resistance to treatment. The same phenomenon, i.e., downregulation of the target molecule, CD73, or a selection for CD73^dim/neg^ cells in the population, could also account for the only transient anti-cancer activity of 22E6 in some treated animals. To prove this hypothesis, we isolated the transplanted PDX cells from the animals at the end of the experiments and analyzed them for CD73 expression by FACS. Intriguingly, reisolated PDX cells from 22E6 treated animals revealed almost no detectable CD73 surface expression and enzymatic activity, whereas cells isolated from control animals maintained the CD73 levels measured prior to transplantation (Figure 7D). Taken together, 22E6 may have a growth delaying effect on CD73 positive ALL cells in vivo, but the leukemia cells quickly became resistant to the treatment by downregulating CD73 or by selection for a CD73^dim/neg^ subpopulation.

## 4. Discussion

CD73 inhibition to interfere with adenosine production is considered a promising way of systemic cancer treatment [23]. To that end, we have generated a novel antibody, 22E6, that blocks CD73 enzyme activity efficiently by steric inhibition or by binding to an allosteric center. Non-competitive inhibition is presumably advantageous in pathological conditions characterized by high local substrate concentrations, such as in the immunosuppressive tumor microenvironment [24,25].

CD73 blocking antibodies described so far inhibit both the membrane-tethered and soluble forms alike. Even though the exact role of soluble CD73 remains to be elucidated, it is known that it is involved in key physiological functions, such as the protection of organs from ischemia, and that it exerts an important immunoregulatory function [26,27,28]. In line, Thompson et al. demonstrated that CD73^−/−^ mice suffered from increased vascular leakage in various organs, which could be partly reversed with soluble CD73 [29]. In this respect, it is thus worth mentioning that 22E6, in contrast to other CD73 blocking antibodies, selectively inhibits membrane tethered CD73 (mCD73), which is overexpressed in many types of cancer, at nanomolar concentrations. It is, therefore, tempting to speculate that 22E6 may result in a lower predisposition to adverse effects in vivo. In addition, similar to RNAi-mediated CD73 suppression [30], 22E6 directly induces apoptosis in TNBC cells and increases their sensitivity to DOX. This observation is in line with that of Serra et al., who described a protective role of ADO in chronic lymphatic leukemia (CLL) cells against apoptosis induced by various chemotherapeutic agents [8].

We not only characterized the 22E6 antibody in various in vitro assays but also evaluated its effect in a first in vivo experiment in immunocompromised NSG mice transplanted with patient derived ALL cells. Unexpectedly, treatment had no or only a very mild effect on tumor burden. To our surprise, reisolated ALL cells from 22E6 treated mice had completely ceased to display CD73 on their surface, whereas those from control animals expressed mCD73 at levels comparable to those prior to transplantation. This observation is proof of the amazing plasticity and resilience of ALL cells and convincingly recapitulates the clinical situation of therapy-induced resistance to treatment. Despite the fact that we did not observe a sustained therapeutic effect of our CD73, we trust that this result is worth publishing because cancer immune escape is also a significant clinical problem and the major cause of the development of resistance to treatment.

Even though CD73 blockade alone revealed no long-lasting antitumor effect in our pilot experiment with PDX ALL cells in vivo, the selection of a CD73^dim/neg^ subpopulation underscores the important role of CD73 for ALL cells. Taken together, we show that the immune checkpoint molecule CD73 is a promising target for specific cancer therapies and that our antibody 22E6 is a novel attractive therapeutic candidate with a new mode of action to be used in combination therapies.

## 5. Patents

A patent application (WO2018215535A1) on the 22E6 antibody has been filed by the Helmholtz Zentrum München in 2017.

## Figures and Tables

**Figure 1 biomedicines-10-00825-f001:**
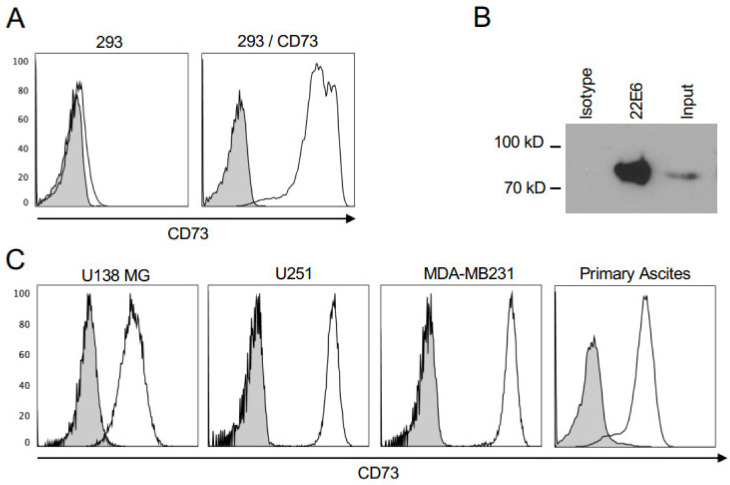
22E6 is CD73-specific. (**A**) 22E6 binds to HEK293 cells stably transfected with a CD73 expression plasmid (293/CD73) but not to parental HEK293 cells (293). (**B**) 22E6 specifically precipitates CD73 from lysates. (**C**) 22E6 binds to various permanent cancer cell lines and cells isolated from primary ascites. Plain histograms in (**A**,**C**) = isotype control antibody.

**Figure 2 biomedicines-10-00825-f002:**
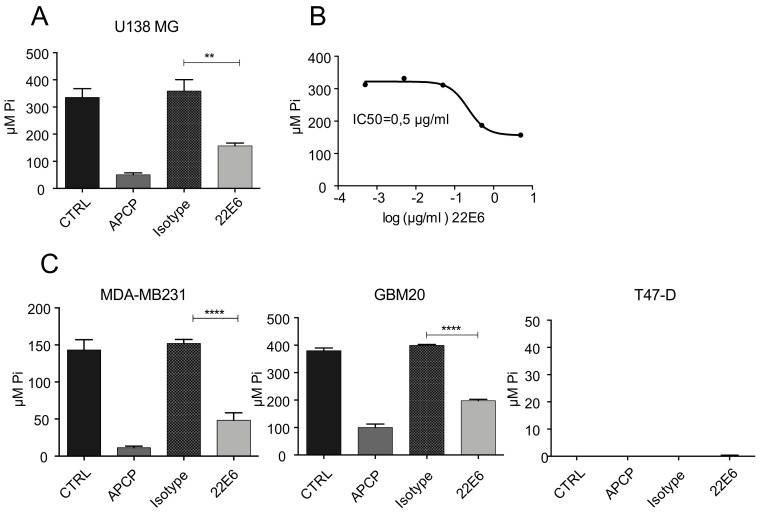
22E6 inhibits CD73 enzyme activity. (**A**) U138 MG glioblastoma cells were incubated with APCP, 22E6 or an isotype control antibody in the presence of AMP. The amount of inorganic phosphate was measured in a Malachite Green assay. (**B**) The IC50 of 22E6 was calculated to be 0.5 µg/mL, corresponding to approximately 3.5 nM. (**C**) The generation of inorganic phosphate from ADP is CD73-dependent and effectively blocked by 22E6 as shown on MDA-MB231 and GBM20 cells. CD73-negative T47-D do not produce detectable amounts of phosphate. Experiments were performed three times. ** *p* < 0.01; **** *p* < 0.0001.

**Figure 3 biomedicines-10-00825-f003:**
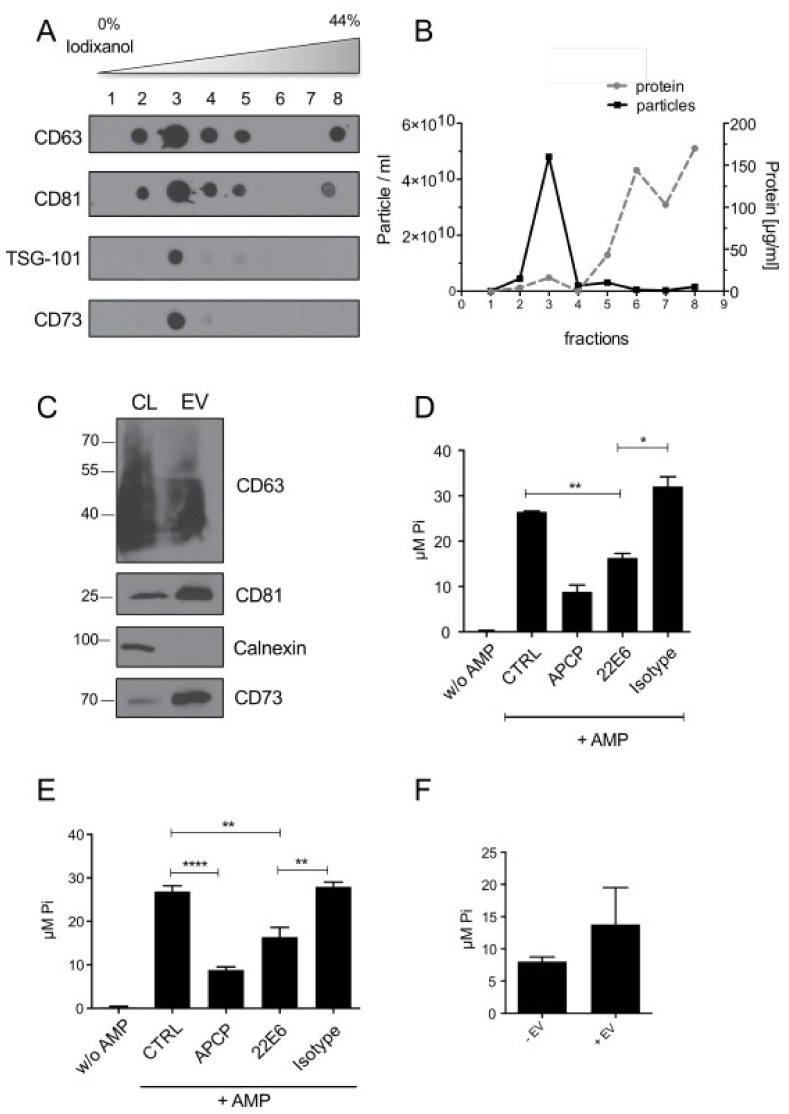
22E6 blocks CD73 enzyme activity on tumor-derived extracellular vesicles. (**A**) TEVs from GBM20 cells were floated into a Optiprep gradient and separated into eight fractions, which were subsequently tested for the presence of the EV markers CD63, CD81 and TSG-101, as well as of CD73. (**B**) Particle numbers in fractions 1 to 8 were counted by NTA, and the protein content was measured in a Bradford assay. (**C**) Fraction 3 contains vesicles which stain positive for CD63, CD81 and CD73, while it is negative for Calnexin indicative for a cell-free preparation. (**D**) APCP and 22E6 block CD73 activity on TEVs from GBM20 cells and (**E**) isolated from primary ascites from a patient with ovarian cancer. (**F**) enzyme activity can be transferred by CD73-positive TEVs from GBM20 cells onto CD73-negative T47-D cells. Experiments were performed at least three times. * *p* < 0.05; ** *p* < 0.01; **** *p* < 0.0001.

**Figure 4 biomedicines-10-00825-f004:**
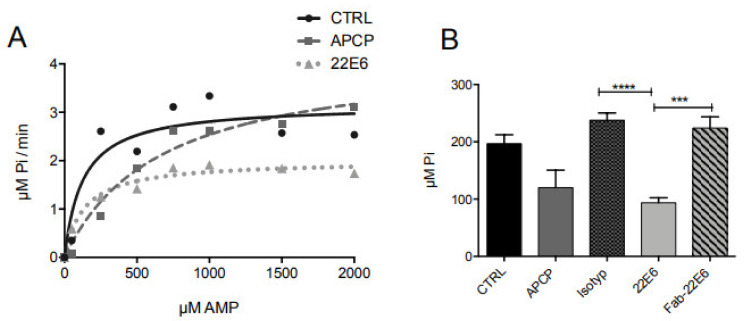
22E6 is a non-competitive inhibitor of CD73. (**A**) Incubation of U138 MG cells with 22E6 and increasing concentrations of AMP resulted in a decreases Vmax. (**B**) The 22E6 antibody but not the 22E6 Fab fragments inhibits CD73 activity. One representative experiment of three is shown. *** *p* < 0.001; **** *p* < 0.0001.

**Figure 5 biomedicines-10-00825-f005:**
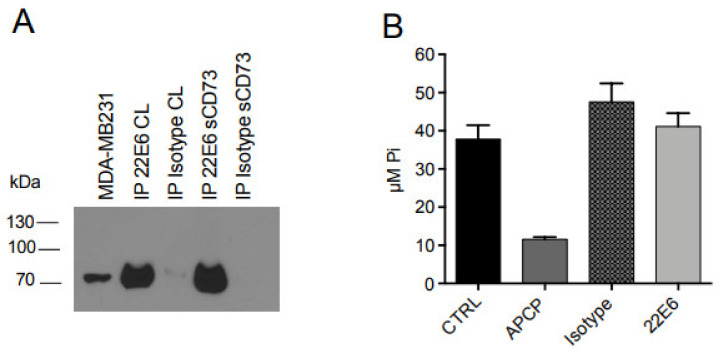
22E6 does not inhibit enzyme activity of soluble CD73. (**A**) specific precipitation of recombinant CD73 by 22E6. (**B**) 22E6 does not detectably inhibit enzyme activity of soluble CD73. Error bars = SD; *p* < 0.001 for APCP, n.s. for Isotype and 22E6. This experiment was performed at least four times.

**Figure 6 biomedicines-10-00825-f006:**
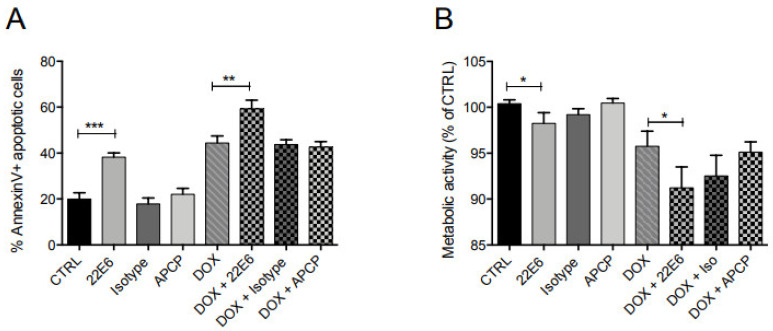
22E6 induces statistically significant apoptosis in triple-negative breast cancer cells. (**A**) 22E6 induces apoptosis in MDA-MB231 TNBC cells alone and in combination with DOX as measured in an Annexin V assay. (**B**) A TMRE assay revealed a reduction of mitochondrial activity in MDA-MB231 cells upon incubation with 22E6 alone or in combination with DOX. One representative experiment of three is shown. * *p* < 0.05; ** *p* < 0.01; *** *p* < 0.001.

**Figure 7 biomedicines-10-00825-f007:**
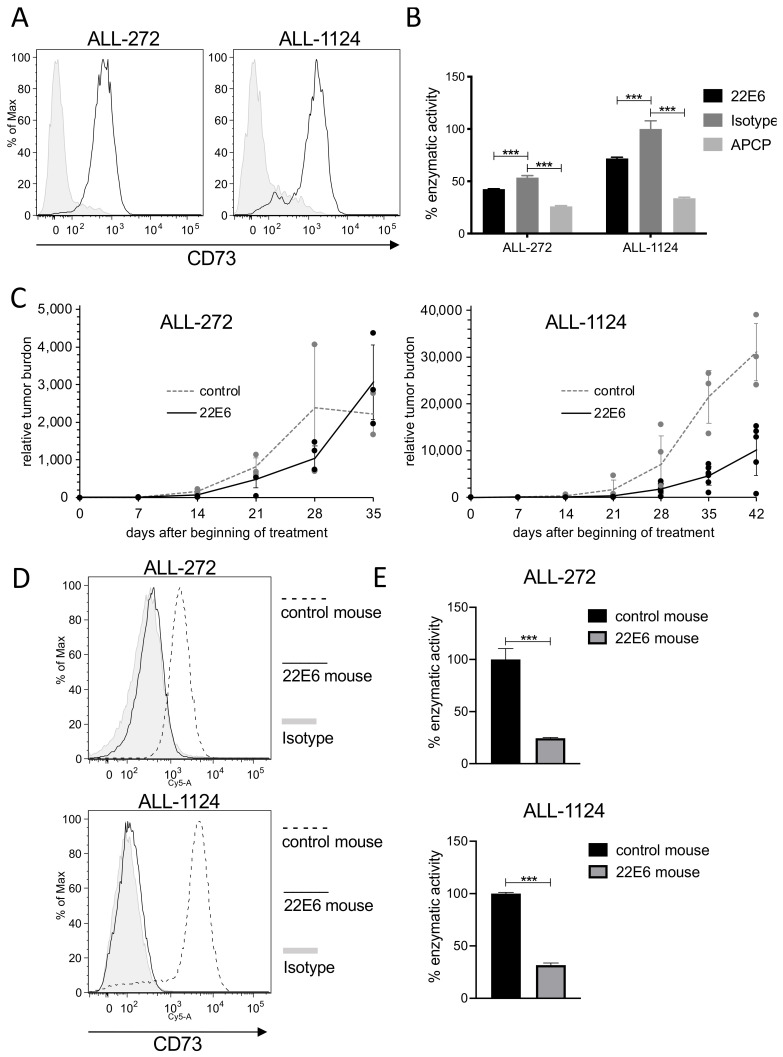
22E6 induces downregulation of CD73 on PDX ALL cells in vivo. (**A**) PDX ALL cells express surface CD73 before transplantation. Black line = 22E6; tinted histogram = isotype control. (**B**) CD73 on PDX ALL cells is enzymatically active, and this activity can be inhibited with 22E6 and APCP. (**C**) NSG mice were transplanted with ALL-272 or ALL-1124 cells. Tumor burden was monitored by bioluminescence imaging (BLI). Mice were treated with 100 µg 22E9 twice per week (*n* = 5 for ALL-272, *n* = 6 for ALL-1124; black triangles) or left untreated (*n* = 3 for ALL-272, *n* = 4 for ALL-1124; grey dots). Tumor burden (BLI signal) relative to treatment start was calculated. Data of two independent experiments were integrated. Shown are individual mice (dots) as well as mean ± standard deviation (line). Tumor burdens of ALL-1124 between control and 22E6 treated mice differed significantly at days 35 (*p* < 0.05) and 42 (*p* < 0.01). At high tumor burden, mice were sacrificed, and cells were isolated. (**D**) Flow Cytometric analysis of isolated xenograft ALL cells from 22E6 treated (solid line) and control mice (dashed line) checking for expression of CD73. Black line = 22E6 anti-CD73; Tinted = control. (**E**) AMPase assay with ALL cell lines testing for enzymatic activity of CD73 of isolated xenograft ALL cells from 22E6 treated and control mice. *** *p* < 0.001.

## Data Availability

Not applicable.

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
