# Peer review of "A Novel Anti-CD73 Antibody That Selectively Inhibits Membrane CD73 Shows Antitumor Activity and Induces Tumor Immune Escape"

_biomedicines, 2022, doi:10.3390/biomedicines10040825_

Round 1
Reviewer 1 Report
Kellner and colleagues have investigated the capacity of a novel antibody to block the efficacy of CD73, which has been described as a pro-tumorigenic membrane-bound enzyme. The targeting of CD73 is of significant interest in the field of tumor immunotherapy and the readers of Biomedicines. The authors provide a comprehensive characterization of their 22E6 antibody, which shows interesting activity. The NSG in vivo experiments are a valuable addition as a proof of principle, although the immune response is obviously lacking. The manuscript is well written and the experiments are set up to investigate the proposed hypotheses. Nonetheless, I do have several concerns that prevent publication at this moment. Thus, I suggest a major revision.
Major concerns:
The last author is the founder of Eximmium, which aims to market the antibody described in this publication. While the patent is mentioned, the role as founder and active participant of Eximmium should be mentioned in the Conflict of Interest.
In all figures:
Disclose how often this experiment is done and whether these data are representative of those other experiments. At least 2 experiments are required to establish reproducibility.
Disclose the error bar that is displayed (include N size if SEM is used) and which statistics are performed.
Figure 1: Please include the description of the negative control in the figure legend (isotype control antibody?).
Provide the omitted data on patient-derived tumor EVs as described in line 306-308, possibly in supplementary material.
Section 3.4: Please provide the data showing comparable binding of the full length 22E6 antibody and the 22E6 FAB fragment.
Figure 5: Please provide statistics and if not statistically significant, mention in the text.
Line 412: The experiment does not test whether the ALL xenograft cells downregulate CD73. A more likely hypothesis is the “selection” in tumor cell expansion that favors CD73dim/neg ALL cells due to 22E6 treatment. Or in other words, apoptosis induces by 22E6 leaves only CD73dim/- ALL cells alive at the end of the experiment.
Line 430: I disagree with the proof that 22E6 aids DOX-induced apoptosis. From the data it shows that the level of apoptosis induced by 22E6 is similar regardless of DOX. One would rather propose a DOX-independent mode of apoptosis induction.
Minor concerns:
The introduction puts the reader on the wrong footing by emphasizing the role of the immune system, which is later not investigated at all. Perhaps less emphasis on immune regulation and more emphasis on current CD73 blocking strategies would be more fitting.
Please refrain from using “immunosuppression” or “immunosuppressive” when this capacity of CD73 is not directly investigated. For example, line 311 and 316. AMP metabolism is not directly analogous to immunosuppression and CD73 is not described as solely immunosuppressive. Also, the authors do not address the immune response in any of their models.
Conceptually, extracellular vesicles are derived from multivesicular bodies in the cell and do not represent the plasma membrane composition. How do the authors discriminate between EV-CD73 and Cell-CD73 in their rationale? Please discuss in the Discussion how the authors expect to use this discrepancy, especially since CD73 is measured on cell membrane (section 3.7) but in the manuscript used in the form of EV-bound CD73.
Figure 1C: Please include T47-D data that is mentioned in the text, but omitted in the figure.
Figure 2C: y-axis is lost in GBM20 figure
Line 294: As it turned out, …
Headings of 3.6 and 3.7 are not correct (22E6).
Line 406: proof = prove
Author Response
Dear Reviewer,
thanks a lot for your constructive comments. We have dealt with them as follows (in bold):
Major concerns:
The last author is the founder of Eximmium, which aims to market the antibody described in this publication. While the patent is mentioned, the role as founder and active participant of Eximmium should be mentioned in the Conflict of Interest. It is correct that I'm a founder of Eximmium, but it is not correct that Eximmium aims to market the antibody described here (or any other CD73 antibody).
In all figures: Disclose how often this experiment is done and whether these data are representative of those other experiments. At least 2 experiments are required to establish reproducibility. This information has been added.
Disclose the error bar that is displayed (include N size if SEM is used) and which statistics are performed. This information has been added.
Figure 1: Please include the description of the negative control in the figure legend (isotype control antibody?). This information has been added.
Provide the omitted data on patient-derived tumor EVs as described in line 306-308, possibly in supplementary material. This has been included as Supplementary figure S1
Section 3.4: Please provide the data showing comparable binding of the full length 22E6 antibody and the 22E6 FAB fragment. Has been included as Supplementary figure S2
Figure 5: Please provide statistics and if not statistically significant, mention in the text. This information has been added.
Line 412: The experiment does not test whether the ALL x enograft cells downregulate CD73. A more likely hypothesis is the “selection” in tumor cell expansion that favors CD73dim/neg ALL cells due to 22E6 treatment. Or in other words, apoptosis induces by 22E6 leaves only CD73dim/- ALL cells alive at the end of the experiment. According to our FACS data, the whole PDX population stained CD73 positive. I therefore think that it is hard to tell whether loss of CD73 in the treated animals is due a loss of CD73 expression or by a selection for a CD73dim/neg subpopulation. I've modified the text accordingly by mentioning either possibility.
Line 430: I disagree with the proof that 22E6 aids DOX-induced apoptosis. From the data it shows that the level of apoptosis induced by 22E6 is similar regardless of DOX. One would rather propose a DOX-independent mode of apoptosis induction. The reviewer is right. The relevant text now reads ' …22E6 alone induced apoptosis and also had a significant additive effect with DOX (Figure 6A)'
Minor concerns:
The introduction puts the reader on the wrong footing by emphasizing the role of the immune system, which is later not investigated at all. Perhaps less emphasis on immune regulation and more emphasis on current CD73 blocking strategies would be more fitting. I partly agree with the reviewer here as CD73 is certainly an important immune modulating enzyme. I agree however that the emphasis was too much on that role. I've therefore deleted one sentence on the consequences of ADO in the immune system and added a short information on the development of clinical development of CD73 blocking strategies. Blocking antibodies and small molecule inhibitors are the two classes of drugs currently investigated by various companies. Detailed information on them are hard to achieve.
Please refrain from using “immunosuppression” or “immunosuppressive” when this capacity of CD73 is not directly investigated. For example, line 311 and 316. AMP metabolism is not directly analogous to immunosuppression and CD73 is not described as solely immunosuppressive. Also, the authors do not address the immune response in any of their models. The reviewer is correct. I've modified the text accordingly, and now only refer to ADO as potentially immunosuppressive.
Conceptually, extracellular vesicles are derived from multivesicular bodies in the cell and do not represent the plasma membrane composition. How do the authors discriminate between EV-CD73 and Cell-CD73 in their rationale? Please discuss in the Discussion how the authors expect to use this discrepancy, especially since CD73 is measured on cell membrane (section 3.7) but in the manuscript used in the form of EV-bound CD73. Cells release different types of vesicles, including exosomes which are derived from multivesicular bodies (MVB) and microvesicles which are derived from the plasma membrane. Following the recommendation of the International Society of Extracellular Vesicles (ISEV), today, EVs are referred to as the mixture of all types of vesicles released from cells. Thus, EVs also include membrane-derived microvesicles.
Figure 1C: Please include T47-D data that is mentioned in the text but omitted in the figure. For the sake of the appearance of Figure 1, I've replaced the histogram of U251 cells (which are not mentioned elsewhere in the manuscript) by a T47-D histogram.
Figure 2C: y-axis is lost in GBM20 figure. This has been corrected
Line 294: As it turned out, … This has been corrected
Headings of 3.6 and 3.7 are not correct (22E6). This has been corrected
Line 406: proof = prove This has been corrected

Reviewer 2 Report
CD73 inhibition to interfere with adenosine production is considered a promising cancer treatment option. This manuscript highlights an important issue. I would only suggest adding the strengths and the limitations of the study with future direction in the discussion.
Author Response
Dear Reviewer,
thanks a lot for your constructive comments. We have dealt with them as follows (in bold):
CD73 inhibition to interfere with adenosine production is considered a promising cancer treatment option. This manuscript highlights an important issue. I would only suggest adding the strengths and the limitations of the study with future direction in the discussion. I've tried to modify the discussion accordingly.
Round 2
Reviewer 1 Report
Clear rebuttal of my concerns by the authors. The manuscript can now be accepted for publication. Best wishes